# MODEL-DRIVEN LABELED DATA FREE FINE-TUNING

## ABSTRACT

Supervised fine-tuning is a prevalent technique for boosting model performance. However, it heavily depends on extensive training over labeled data. This paper introduces a novel model-driven fine-tuning method that operates independently of supervised training and labeled data. By harnessing the collective intelligence of a diverse model pool, our method enhances individual model performance through a two-phase process. Initially, we consolidate the expertise of the models within the pool to create a general meta-model. This meta-model then serves as a guide for iteratively fine-tuning the original models in a few shots, promoting a synergistic improvement in performance. Our experimental results show that this model-driven approach not only surpasses the performance of full-parameter fine-tuning models but also does so without the need for supervised training. This breakthrough offers a cost-effective and scalable alternative to traditional supervised fine-tuning, addressing the challenge of data scarcity and paving the way for future research in unsupervised model enhancement. Our work represents a significant step towards making fine-tuning techniques more accessible and practical in environments where labeled data is limited or even unavailable.

## 1 INTRODUCTION

Foundation models in computer vision (CV) and natural language processing (NLP) have seen unprecedented development, driven by the flourishing growth of data and computational power. The surge of foundation models can be largely attributed to their ability to capture intricate patterns and relationships within the ever-expanding data. Notable examples include BERT Devlin et al. (2018) in the field of NLP and Vision Transformer Dosovitskiy et al. (2020) in CV. The future trajectory of foundation models, spanning from vision Zhai et al. (2022) and language Hoffmann et al. (2022) to multi-modal Aghajanyan et al. (2023) contexts, is likely to inexorably involve the continual scaling of model sizes and the innovation of network architectures. A primary obstacle in developing foundational models is the high cost of training, hindered by time-consuming processes and the demand for high-performance computing resources. The sheer scale of these models necessitates the use of advanced hardware, such as GPUs and TPUs. Then, the extensive need for large volumes of labeled data further compounds the challenges in model development. Collecting, curating, and annotating vast amounts of data is a labor-intensive and resource-intensive task.

Fine-tuning has become a prevalent strategy to address the prohibitive expenses of foundational model training. This approach leverages pre-existing models and adjusts them to better suit specific tasks, reducing the need for extensive training and computational resources. Full fine-tuning (FFT) involves updating all the parameters of a pre-trained model, allowing the model to fully adapt to the new task, potentially achieving high performance. Parameter-efficient fine-tuning (PEFT) has been introduced to further reduce computational expenses. PEFT concentrates on updating a limited subset of parameters or incorporating additional lightweight modules, thereby achieving efficiency gains. Notably, Low-Rank Adaptation (LoRA) Hu et al. (2021) stands out as a prevailing approach that fine-tunes large language models by introducing low-rank matrices to the existing weight matrices, allowing for significant reductions in the number of trainable parameters. This method not only reduces the computational burden but also minimizes the risk of overfitting.

Although fine-tuning provides substantial advantages, it still demands costly training. Whether executed through FFT or PEFT, the fine-tuning process inherently incurs computational overhead. This is due to the need to update and optimize model parameters for tailored tasks. Moreover, even with PEFT methods like LoRA, which aims to reduce the number of trainable parameters, the need

for labeled data remains a critical bottleneck. Labeled data is essential for fine-tuning models, as it provides the necessary supervision to guide the learning process. In domains like medicine Kebaili et al. (2023), low-resource language Magueresse et al. (2020); Ranathunga et al. (2023), and rare objection detection Wang et al. (2020b); Minderer et al. (2024), high-quality labeled data is scarce, further complicating the fine-tuning process and limiting the applicability of these techniques.

Unsupervised fine-tuning refers to fine-tuning using unlabeled data. This learning approach leverages the intrinsic structure of the data to infer patterns, relationships, and representations without the guidance of explicit annotations Jaiswal et al. (2020); Liu et al. (2021). The primary challenge of unsupervised fine-tuning lies in the ambiguity of the optimization direction and the design of the loss function Fang et al. (2024). It is difficult to evaluate the performance of unsupervised models due to the absence of ground truth labels. This makes it hard to assess the quality and applicability of the learned representations. Additionally, unsupervised models may struggle to capture relevant features for specific tasks, resulting in suboptimal performance compared to the supervised schemes Heckler et al. (2023).

In this paper, we propose a novel unsupervised fine-tuning method known as model-driven fine-tuning, which harnesses existing models without needing labeled data. Two stages are included in our proposed method. First, we propose a novel method for meta-model construction leveraging pre-existing fine-tuned models within the model pool using an unsupervised approach. Specifically, we propose an effective unsupervised learning method based on information entropy, which addresses the issues of entropy minimization's error accumulation and sensitivity to harmful samples. This method successfully synthesizes a general meta-model, achieving efficient versatility. Secondly, we refine the meta-model to enhance its specialization. This is achieved by conducting an unsupervised adaptation to minimize the disparity in representations between the meta-model and the original fine-tuned model. Experimental results demonstrate that the meta-model can be effectively fine-tuned with just several dozen steps of representation alignment, surpassing the performance of models fine-tuned with full parameters, all without the involvement of any labeled data.

**Contributions.** (1) We introduce a pioneering unsupervised fine-tuning method termed model-driven fine-tuning. This framework innovatively utilizes pre-existing trained models to perform fine-tuning tasks without relying on labeled data, thereby overcoming the limitations of traditional supervised learning methods. (2) We propose a novel loss function designed specifically for unsupervised fine-tuning. This loss function is grounded in information entropy principles, adeptly addressing challenges of error accumulation in entropy minimization. It can be used to build a general model in an unsupervised way. (3) Experimental results demonstrate that our method matches and even exceeds the performance of full-parameter fine-tuning without labeled data. Furthermore, in a multi-task learning context, our model-driven approach has been shown to outperform the state-of-the-art models, underscoring its superiority in handling complex, real-world scenarios without the need for labeled data.

## 2 RELATED WORKS

**Supervised Fine-tuning.** Supervised Fine-tuning has emerged as a pivotal technique for adapting pre-trained language models to specific tasks or domains. This allows the model to specialize and enhance its performance on that task while retaining the broad knowledge it gained during pre-training. Full Parameter Fine-tuning (FFT) involves updating all the parameters of the pre-trained model. This approach can lead to significant performance improvements as it allows the model to fully adapt to the new task. However, it is computationally expensive and requires much labeled data. FFT is beneficial when there are sufficient data and computational resources. Still, it may not be practical for all the applications due to its high cost and the risk of overfitting when limited data is available. Parameter-efficient fine-tuning methods, on the other hand, fine-tune only a small subset of the model's parameters. This approach is designed to be more resource-efficient and to reduce the risk of catastrophic forgetting, where the model loses previously learned knowledge. PEFT can be classified into four main categories: additive fine-tuning, partial fine-tuning, reparameterized fine-tuning, and hybrid fine-tuning, as detailed in Xu et al. (2023). Within these categories, classic methods include Adapter Houlsby et al. (2019), Prompt-tuning Lester et al. (2021), BitFit Zaken et al. (2021), and FISH MASK Sung et al. (2021) for additive fine-tuning; LoRA Hu et al. (2021)

and QLoRA Dettmers et al. (2024) for reparameterized fine-tuning; and UniPELT Mao et al. (2021) and AutoPEFT Zhou et al. (2024) for hybrid fine-tuning.

**Unsupervised Fine-tuning.** Unsupervised fine-tuning Huang et al. (2020); Tanwisuth et al. (2023) is a distinctive technique that enhances model performance by integrating both labeled and unlabeled data. Self-supervised learning (SSL) Hinton & Salakhutdinov (2006) trains models to predict aspects of the input data itself, without relying on external labels. While it effectively utilizes redundant unlabeled data, SSL faces challenges in learning imperfect representations and necessitates the meticulous design of pretext tasks Gidaris et al. (2018). The entropy-minimization method adjusts the model based on the current information entropy, offering simplicity but suffering from error accumulation and potential degradation after numerous optimization steps. Contrastive learning Wang & Qi (2022); Jaiswal et al. (2020), a prevalent SSL method, focuses on learning representations where similar examples are closely grouped and dissimilar examples are widely separated in the embedding space. This approach yields robust features but is sensitive to data augmentation choices and demands substantial computational resources Wu et al. (2024). Cluster-based methods Zhou et al. (2003) group unlabeled data into clusters based on data distribution, using these groupings as pseudo-labels to guide the learning process. Although naive and capable of revealing inherent data structures, these methods are sensitive to the choice of distance metrics Yang et al. (2016). None of the aforementioned unsupervised fine-tuning methods are capable of fine-tuning and enhancing model performance with limited unlabeled data.

# 3 METHOD

## 3.1 OVERVIEW

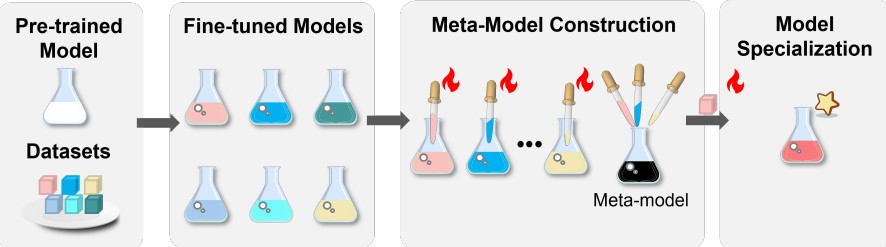

Figure 1: Overview of our proposed method. Two stages are included. (1) To harness the existing models within the model pool to construct a general meta-model with the same size, in an unsupervised manner. (2) To undertake the few-shot model-driven fine-tuning facilitating specialization from the general meta-model.

The complete architecture of our model-driven fine-tuning method is illustrated in Figure 1. It consists of two primary stages. Initially, we harness the existing models within the model pool to construct a general meta-model with the same size, in the unsupervised manner. The meta-model is initialized leveraging fine-tuned models with constant indices, in the granularity of the task-specific or layer-specific. Subsequently, we introduce an innovative unsupervised optimization technique to refine the layer-specific indices. The meta-model is effectively built with the optimized indices, having assimilated the versatile capability from the constituent models. In the second stage, we undertake the model-driven fine-tuning facilitating specialization from the general meta-model. Precisely, we fine-tune the meta-model by aligning its representations with those of a pre-existing fine-tuned model, thereby enabling a tailored specialization.

## 3.2 META-MODEL CONSTRUCTION

**Meta-model Initiation.** To build a general meta-model without supervised training, we leverage the pre-existing fine-tuned models, which have the same architecture but are fine-tuned on different datasets, to build our initialized meta-model. Given the $N$ pre-existing fine-tuned models $\{\theta_k\}_1^N$ and the pre-trained model $\theta_{pre}$, we extract the task vector of model $\theta_k$ by weight subtraction as

$\tau_k = \theta_k - \theta_{pre}$, following the task arithmetic method Ilharco et al. (2023). Then, using the constant indices $\lambda_k$ for each task vector, the meta-model $\theta^M$ is initialized following the below equation:

$$\theta^M = \theta_{pre} + \sum_{k=1}^{N} \lambda_k \cdot \Phi(\tau_k) \tag{1}$$

where $\lambda_k$ indices are the optimization objectives to be refined, which are initialized using constant values. Indices $\lambda_k$ dictate the degree to which the $k$-th model contributes to the meta-model. $\Phi(\tau_k)$ represents the transformation method applied to the task vector $\tau_k$. When $\Phi(\tau_k) = \tau_k$, Eqn. 1 corresponds to the standard task arithmetic method. Due to weight conflicts and not increasing the mode capacity, Eqn. 1 results in significant performance degradation between the individual fine-tuned models and the meta-model. However, the meta-model has nonetheless attained an initial multitasking capability, which denotes the degree of versatility.

**Analysis on Standard Entropy Minimization.** Entropy minimization (EM) Wang et al. (2020a) has emerged as a self-supervision technique grounded in the hypothesis that a well-calibrated model should assign low entropy to its predictions. This approach leverages entropy as a surrogate label to facilitate self-supervision. We employ standard EM to refine model indices, initializing them at 0.3 Yang et al. (2023); Ilharco et al. (2023). The outcomes are detailed in Table 1, where *classes Spear. rho* represents the Spearman correlation coefficients between entropy and actual loss, and *Drop Ration* signifies the performance degradation between individual models and the meta-model with learned indices. Table 1 reveals that datasets with a larger number of classes typically exhibit more pronounced performance degradation and lower Spearman correlation coefficients. This is due to the naive EM algorithm's failure to account for diverse class cardinalities, treating each task uniformly and resulting in suboptimal Spearman correlation coefficients. Furthermore, EM introduces randomness in the initial optimization stage, and there is no mechanism to rectify potential errors in subsequent optimization steps, leading to severe error accumulation Press et al. (2024). As a result, EM cannot fully serve as a proxy for guiding the optimization of model indices.

Table 1: Comparative Analysis of Datasets and Entropy Minimization Performance.

|  | **High Class Cardinality** | | | | | **Low Class Cardinality** | | |
|---|---|---|---|---|---|---|---|---|
|  | *SUN397* | *Cars* | *DTD* | *RESISC45* | *GTSRB* | *SVHN* | *EuroSAT* | *MNIST* |
| *Classes* | 397 | 196 | 47 | 45 | 43 | 10 | 10 | 10 |
| *Spear. rho* | 0.61 | 0.67 | 0.70 | 0.79 | 0.91 | 0.92 | 0.86 | 0.98 |
| *Drop Ratio* | **16.7**% | **21.2**% | **30.9**% | **28.5**% | **20.6**% | 15.3% | 15.9% | 5.3% |

**Debiasd Entropy Minimization.** We propose a novel unsupervised optimization method based on EM, which is more suitable to refine the model indices in Eqn. 1, in order to mitigate the performance degradation and improve the capability of meta-model on multi-tasking. Specifically, we design a joint loss function that harmoniously integrates self-supervision and cooperative supervision. We harness the cross-information between the meta-model and individual models to aid the self-supervised optimization. In the initial stage, the optimization is no longer ambiguous, thus eliminating randomness and mitigating error accumulation. At each step, the self-entropy loss focuses on local supervision and information, while the cooperative loss considers global guidance. Inspired by knowledge distillation techniques Xu et al. (2024), this joint loss effectively mitigates bias and directs the update of multiple model indices, ensuring a balanced and comprehensive optimization process. Concretely, we have crafted methods with two distinct levels of granularity.

**a) Task-specific.** We view the entire model holistically and assign identical indices to each model parameter. For each input of task $k$, the task-specific loss is denoted as follows:

$$\mathcal{L}(x_k; \lambda_k) = \alpha \cdot \left( -\sum_{k=1}^{K} y_k(x_k) \cdot \log \theta_M(x_k; \lambda_k) \right) + H(\theta_M(x_k; \lambda_k)) \tag{2}$$

where $\theta_M(x_k; \lambda_k)$ is the output of the meta-model with current indices $\lambda_k$; $H(\cdot)$ is the information entropy function; $y_{(k)}$ is the output of the $k$-th model; $\alpha$ is a hyperparameter to control the impact from cooperative supervision.

**b) Layer-specific.** Neural networks typically extract features at varying levels of abstraction across different layers. We employ Center Kernel Alignment (CKA) Kornblith et al. (2019) to visualize the layer-wise similarity among each compact model. Notably, when compared to the same model EuroSAT, different models exhibit varying degrees of similarity at different layers 2. Deeper layers generally show greater divergence, whereas shallower layers, except the first layer, tend to be more similar. More visualization results are shown in the Appendix 8. As a result, the overall model's unified indices are too sparse. To address this, we refine the layer-specific debiased entropy minimization approach to operate at the layer level. The formulation is as follows:

$$\mathcal{L}(x_k; \lambda_k^l) = \alpha \cdot \left( - \sum_{k=1}^{K} \sum_{l=1}^{L} y_k^l \cdot \log \theta_s(x_k; \lambda_k^l) \right) + H(\theta_M(x_k; \lambda_k^l)) \tag{3}$$

where $\lambda_k^l$ signifies the layer-specific indices for layer $l$ of the $k$-th cooperative model.

**Class Cardinality-based Sample Filtering.** Standard EM necessitates a substantial number of unlabeled samples, and higher-entropy samples may impede the optimization process Grandvalet & Bengio (2004). Adapting models with samples exhibiting extremely high entropy can degrade performance, whereas low-entropy samples markedly enhance model performance Niu et al. (2022). Consequently, we aim to simultaneously reduce the unsupervised optimization cost and decrease model uncertainty caused by high-entropy samples. Given that class cardinality influences the optimization of model indices, we propose to filter samples while considering both sample entropy and class cardinality. Specifically, we introduce a class cardinality-based sample filtering method, denoted as $F_{ent}(x; \theta_i)$. This method only considers samples with entropy below a threshold specific to their class cardinality for optimization, given by:

$$F_{ent}(x; \theta_k) = \mathbb{I}_{H(x;\theta_k) < H_0(k)}(x; \theta_k), \quad H_0(i) = \mu \cdot \log C_k \tag{4}$$

where $\mathbb{I}$ denotes the indicator function that constructs the entropy filtering function $F_{ent}$; $H_0^k$ represents the filtering threshold about class cardinality $C_k$ of task $k$. To standardize the entropy within a batch, the following coefficients are computed:

$$\gamma^{\theta_k} = \frac{1}{\exp[H(x; \theta_k) - H_0^k]} \cdot \frac{B}{B - N_{F_{ent}}}, \tag{5}$$

where $B$ denotes the batch size and $N_{F_{ent}}$ denotes the number of unfiltered samples in the current batch. Integrating Eqn. 4, Eqn. 5, and either Eqn. 2 or Eqn. 3, we formulate our comprehensive objective function for either task-specific construction (Eqn. 2) or layer-specific construction (Eqn. 3), as follows:

$$\min_{\lambda_k} \gamma^{\theta_k} \cdot F_{ent}(x; \theta_k) \cdot \mathcal{L}(x_k; \lambda_k) \tag{6}$$

By optimizing $\lambda_k$ in Equation 6, we can derive task-specific or layer-specific indices. Using the learned indices, combined with Eqn. 1, the meta-model is constructed. The algorithmic process for meta-model constructing is outlined in Algorithm 1.

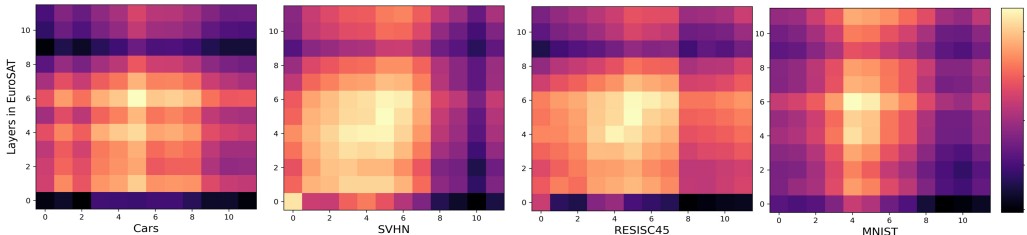

Figure 2: Layer-specific similarity between models calculated by CKA. Light colors indicate high similarity, and vice versa.

### 3.3 MODEL SPECIALIZATION

With the meta-model $\theta^M$ established, we proceed to the second stage, where model-driven fine-tuning is implemented. The meta-model serves as a versatile foundation, demonstrating strong performance in multi-tasking (refer to Section 4.2). This indicates that the meta-model has effectively assimilated capability from a diverse range of fine-tuned models.

**Meta-model Specialization.** Given meta-model $\theta^M$ with strong versatility, we subsequently aim to specialize this model toward the specific task, facilitating the fine-tuning process. Specifically, we achieve model specialization by leveraging representation alignment Sucholutsky et al. (2023). In detail, we extract the existing fine-tuned model from the model pool that corresponds to the task we wish to fine-tune. We then guide the meta-model's representation to re-adapt towards this tuned model for a few steps, thereby inducing a shift in the meta-model's representation. Specifically, we minimize the L1 distance between their representations and subsequently update the optimization of the meta-model's indices. The formula is as follows:

$$min\mathcal{L}_{L1}(y(\theta^M), y(\theta^k)), \tag{7}$$

where $y(\cdot)$ represents the representation of the final layer and $\theta^k$ denotes the aligned model. By performing 20 steps of minimization on Equation 7, the model achieves specialization, thereby efficiently completing model-driven fine-tuning.

### 3.4 MODEL-DRIVEN FINE-TUNING ALGORITHM

The algorithm is concisely presented as Algorithm 1. For each task, the algorithm accepts the following inputs: (1) the pre-trained model $\theta_{pre}$; (2) the teacher models $\{\theta_t^1, \theta_t^2, ..., \theta_t^N\}$, and (3) the task inputs $\{I_1, I_2, ..., I_N\}$. Firstly, we undertake some preliminary steps: acquiring task vectors and initializing the meta-model (*line 3-4*). At each iteration, we calculate the loss for each task (*line 6-10*) and subsequently derive the cumulative loss by summing these values (*line 11*). Based on the aggregate loss, we adjust the model indices corresponding to each task and subsequently update the meta-model parameters $\theta_s$ (*line 11-12*).

---

**Algorithm 1** Model-Driven Fine-tuning

---

1: **Input**: $K$ fine-tuned models $\{\theta_t^1, \theta_t^2, ..., \theta_t^K\}$, default indices $\{\lambda_i\}_1^K$, unlabeled samples $\{D_i\}_1^K$
2: **Output**: the fine-tuned model $\theta^*$
3: Extract task vector$\tau_k = \theta_t^k - \theta_{pre}$    $k = 1, 2, ..., N$
4: Initialize Meta-model following Equation 1
5: **for** $i = 1, ..., T$ **do**
6:    **for** $k = 1, ..., K$ **do**
7:        Sample a batch of data $(x_k^i)$ from $D_k$
8:        Filter data following Equation 4
9:        Reweigh data following Equation 5
10:        Compute loss $\mathcal{L}(x_k^i; \lambda_k)$ following Equation 2
11:    **end for**
12:    $\mathcal{L}_i = \sum_{k=1}^N \mathcal{L}(x_k^i; \lambda_k)$
13:    $\lambda_k^{(i)} \leftarrow \lambda_k^{(i-1)} - \nabla \mathcal{L}_i.$
14:    Update meta-model $\theta_s$ following Equation 1
15: **end for**
16: Sample few batches of data $(x^*)$ from $D_*$
17: $y(\theta^M), y(\theta^*) \leftarrow$ inference $(x_*)$
18: Minimize $\mathcal{L}_{L1}$ following Equation 2
19: Update indices $\{\lambda_i\}_1^K$ following Equation 7
20: $\theta^* \leftarrow \theta_M$

---

## 4 EXPERIMENTS

**Datasets and Models.** We assess the efficacy of our proposed framework across eight diverse image classification datasets, varying in class cardinalities. These include SUN397 Xiao et al. (2016), Cars Krause et al. (2013), RESISC45 Cheng et al. (2017), EuroSAT Helber et al. (2019), SVHN Netzer et al. (2011), GTSRB Stallkamp et al. (2011), MNIST LeCun (1998), and DTD Cimpoi et al. (2014). Our backbone and pre-trained models are the ViT-base with a patch size of 32x32 and ViT-Large with a patch size of 14x14 Dosovitskiy et al. (2020) from CLIP Radford et al. (2021) models. The fine-tuned models are sourced from a publicly accessible hub [1].

---

[1]https://github.com/mlfoundations/task_vectors

Table 2: Fine-tuning performance comparison

| Method | SUN397 | Cars | RESI45 | EurSAT | SVHN | GTSRB | MNIST | DTD | Avg. |
|---|---|---|---|---|---|---|---|---|---|
| ViT-B/32 | | | | | | | | | |
| Pre-Trained Dosovitskiy et al. (2020) | 63.2 | 59.6 | 60.2 | 45.0 | 31.6 | 32.6 | 48.3 | 44.4 | 48.1 |
| Multi-Task Learning Huang et al. (2024) | 73.9 | 74.4 | 93.9 | 98.2 | 95.8 | **98.9** | 99.5 | 77.9 | 88.9 |
| Full Fine-tuning Ilharco et al. (2023) | 75.3 | 77.7 | **96.1** | 99.7 | **97.5** | 98.7 | **99.7** | 79.4 | 90.5 |
| **Ours** | **79.3** | **78.4** | **96.1** | **99.8** | **97.5** | 98.8 | 99.5 | **79.6** | **91.1** |
| ViT-L/14 | | | | | | | | | |
| Pre-Trained Dosovitskiy et al. (2020) | 66.8 | 77.7 | 71.0 | 59.9 | 58.4 | 50.5 | 76.3 | 55.3 | 64.5 |
| Multi-Task Learning Huang et al. (2024) | 80.8 | 90.6 | 96.3 | 96.3 | 97.6 | 99.1 | 99.6 | 84.4 | 93.5 |
| Full Fine-tuning Ilharco et al. (2023) | 82.3 | 92.4 | 97.4 | **100** | 98.1 | 99.2 | 99.7 | 84.1 | 94.2 |
| **Ours** | **84.6** | **92.7** | **97.4** | 99.7 | **98.1** | **99.4** | **99.7** | **84.8** | **94.6** |

**Compared Methods and Implementation.** We assess our proposed framework by comparing its performance against both the pre-trained model and their fully fine-tuned models. Then, to further evaluate the performance of the meta-model on multi-tasking, the multi-task learning model is compared, which is trained by supervised training. In the first stage of our framework, the model indices $\lambda_k$ are equally initialized as 0.3 Ilharco et al. (2023). In the second stage, 20 steps are executed, and accuracy serves as the metric for evaluating all tasks.

### 4.1 COMPARISON OF FINE-TUNING PERFORMANCE AND EFFICIENCY

**Performance Comparison.** We have conducted a thorough analysis of fine-tuning performance across full fine-tuning benchmarks for the ViT-B/32 and ViT-L/14 models, as delineated in Table 2. Our model-driven fine-tuning method referred to as *Ours*, has been compared with Pre-Trained and Full Fine-tuning approaches, as well as Multi-Task Learning. The results are shown in Table 2. According to Table 2, several observations can be made:

(i) Both on the ViT-B/32 and ViT-L/14 models, our method achieves the highest average accuracy. Across both models, our method outperforms the full fine-tuning method in 7 out of the 8 datasets, without the need for labeled data or extensive supervised training.

(ii) On the SUN397 dataset, our method attains 79.3% for ViT-B/32 and 84.6% for ViT-L/14, representing significant improvements over the next best methods. Notably, our method demonstrates particular strength on the EurSAT dataset, achieving perfect or near-perfect scores for both model sizes. This indicates that our fine-tuning approach is highly effective for certain complex tasks.

(iii) For the MNIST dataset with ViT-B/32 and EuroSAT with ViT-L/14, our method lags behind by only 0.2% and 0.3%, respectively, as these are simple tasks that have been extensively trained. We infer that, on these two datasets, the full fine-tuned models have likely overfit. In conclusion, our fine-tuning method has demonstrated greater effectiveness in labeled-free fine-tuning, providing substantial improvements over existing approaches, particularly for larger models like ViT-L/14. This underscores the effectiveness of our approach in enhancing model performance across a diverse range of tasks.

**Efficiency.** We further assess the efficiency of our proposed method based on the ViT-B/32 model. Figure 3 illustrates the performance variance across optimization steps. In Figure 3, the red dashed line represents the full fine-tuning baseline. As depicted, our method converges on all datasets within 80 steps, and notably, it achieves convergence in just 20 steps for MNIST and 30 steps for EuroSAT, respectively, thereby emphatically demonstrating the efficiency of our proposed method. Supplementary analysis for the ViT-L/14 model is elaborated upon in Appendix B.

Further, we have meticulously assessed the efficiency and the utilization of trainable parameters, as depicted in Table 3. The findings indicate that our proposed method markedly surpasses the established baseline with respect to optimization time and trainable parameters Key insights include: (i) Our method exhibits an impressive speedup ratio that surpasses 9 times for the ViT-B/32 model, and this ratio increases with the model size. Specifically, for the ViT-L/14 model, the speedup ratio is even more pronounced. (ii) Our method demands a substantially lower number of trainable parameters compared to the FFT method, utilizing only 3.1% and 2.1% of the parameters for the ViT-B/32 and ViT-L/14 models, respectively. This reduction is because only the model indices need

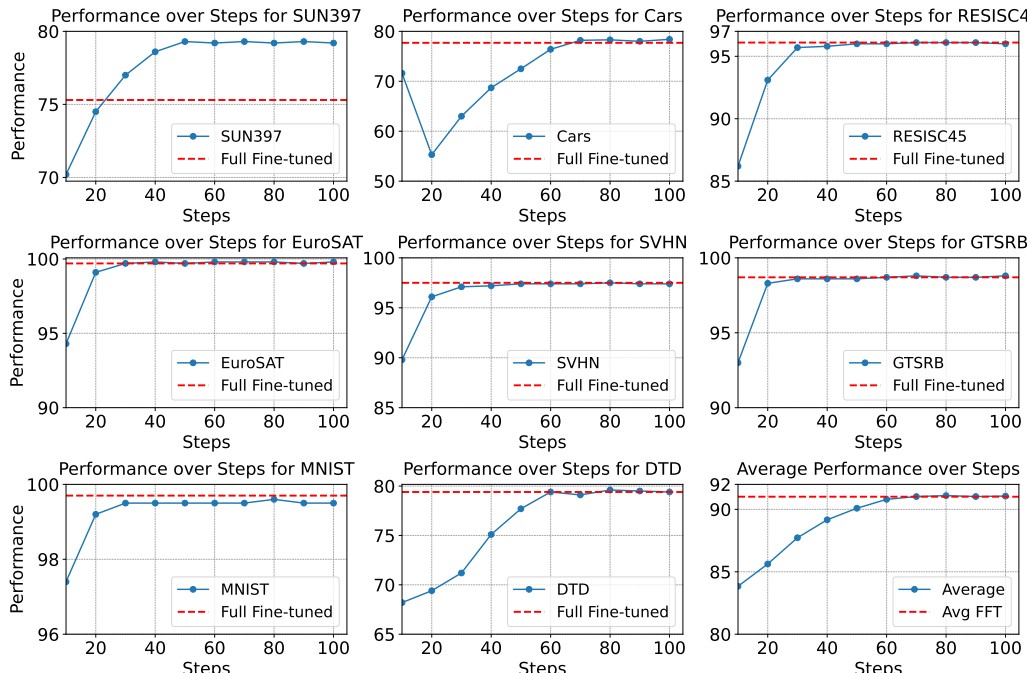

Figure 3: Performance versus steps compared to full fine-tuning based on ViT-B/32. Each red dashed line signifies the performance of full fine-tuning.

to be optimized, thanks to our innovative layer-specific minimization approach. which is linearly proportional to the number of layers within the model, as detailed in 3. In contrast, the trainable number of the FFT method is associated with the entire model parameter size.

Table 3: Efficiency Comparison

| Model | Method | Avg Optim. Time(s) | Speedup Ratio | Trainable Params | Rel. |
|-------|--------|---------------------|---------------|------------------|------|
| ViT-B/32 | FFT | 361.9 (±50) | 1.0 | 108.2MB | 100% |
| | Ours | 37.8 (±5.0) | **9.57 ↑** | 3.4MB | **3.1%** |
| ViT-L/14 | FFT | 3300 (±200) | 1.0 | 326.7 MB | 100% |
| | Ours | 108 (±17) | **30.6↑** | 6.8MB | **2.1%** |

## 4.2 ABLATION STUDIES

**Ablation for Meta-model Construction.** We conduct an ablation study on the effectiveness of meta-model construction. Given the challenge of quantifying the degree of model versatility, we employ the model's performance on multi-task scenarios as a proxy, specifically using average accuracy. We compare a range of unsupervised model integration methods without increasing model parameters, categorizing them into direct weight manipulation approaches and task vector-based strategies. The former includes Model Soups Wortsman et al. (2022), Fisher Merging Matena & Raffel (2022), and RegMean Jin et al. (2022). The latter encompasses Task Arithmetic Ilharco et al. (2023), Ties-Merging Yadav et al. (2023), and AdaMerging Yang et al. (2023). All experimental settings are standardized, encompassing identical networks and datasets. Figure 4 illustrates that our proposed method attains the highest average performance across all evaluated methods, thereby substantiating the effectiveness of our method in building versatile meta-models.

**Ablation on Class Cardinality-based Sample Filtering.** To demonstrate the efficacy of class cardinality-based sample filtering, we have compared the performance achieved with and without employing this technique. Utilizing an identical experimental setup, with the sole variation being

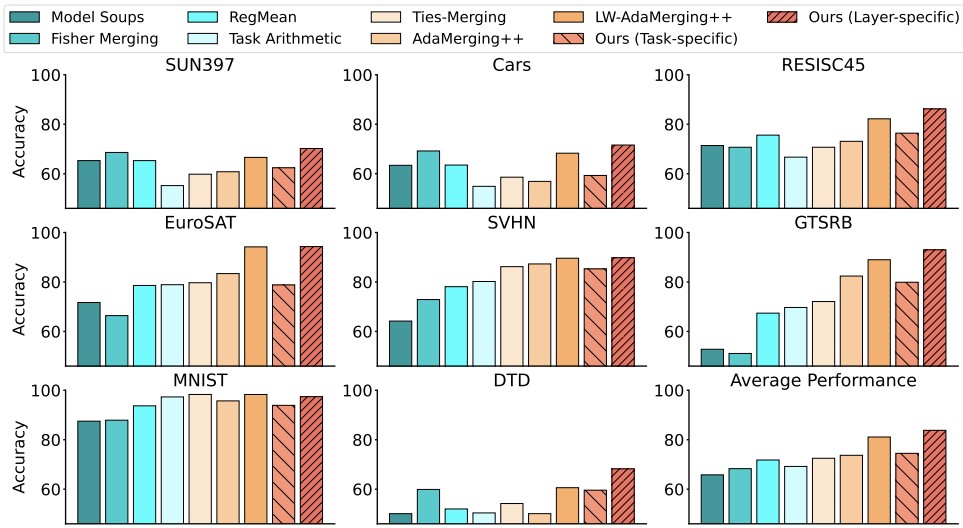

Figure 4: Ablation for meta-model construction methods.

the data refinement process illustrated in Figure 5, the results conclusively indicate that our filtering method outperforms the non-filtered approach regarding overall average performance. Moreover, our method exhibits a faster convergence rate than the naive method.

**Ablation for Optimization Function.** To clarify the effectiveness of debiased entropy minimization, we conducted an ablation study on the optimization function. Specifically, we compared the performance of joint supervision, singular self-supervision, and singular cooperated supervision during the optimization process. Figure 6 demonstrates that our proposed loss function achieves the highest performance across all tasks. Additionally, $L_{CE}$ and $L_H$ exhibit varying levels of superiority on different datasets. The experimental results unequivocally demonstrate the effectiveness of our method.

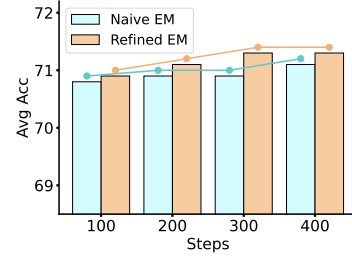

Figure 5: Ablation for sample filtering.

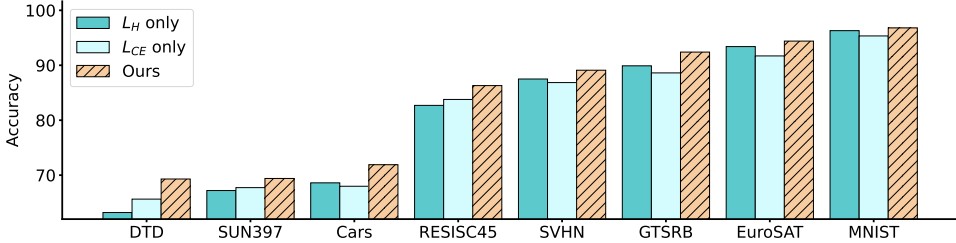

Figure 6: Ablation for optimization function.

## 5 CONCLUSION

Supervised fine-tuning is computationally expensive and often necessitates much labeled data, whereas unsupervised fine-tuning faces challenges in providing effective guidance and designing appropriate supervision. In this paper, we introduce a novel model-driven labeled free fine-tuning method, which allows fine-tuning and boosting models without labeled data and extensive training. Leveraging the collective intelligence of diverse models, our two-phase approach significantly

enhances individual model performance by creating a general meta-model and subsequently fine-tuning the original model in a few shots. Our method, which operates independently of any supervised data, surpasses the performance of supervised full-parameter fine-tuning. Our experimental results demonstrate that this model-driven approach offers a cost-effective and scalable alternative to traditional supervised fine-tuning. It effectively addresses the challenge of data scarcity, making fine-tuning techniques more accessible and practical in environments where labeled data is limited or inaccessible. This breakthrough paves the way for future research in unsupervised fine-tuning.

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

# APPENDIX

## A  SIMILARITY

The supplementary layer-specific similarity analysis is presented. The CKA Kornblith et al. (2019) visualization about MLP layers is as follows:

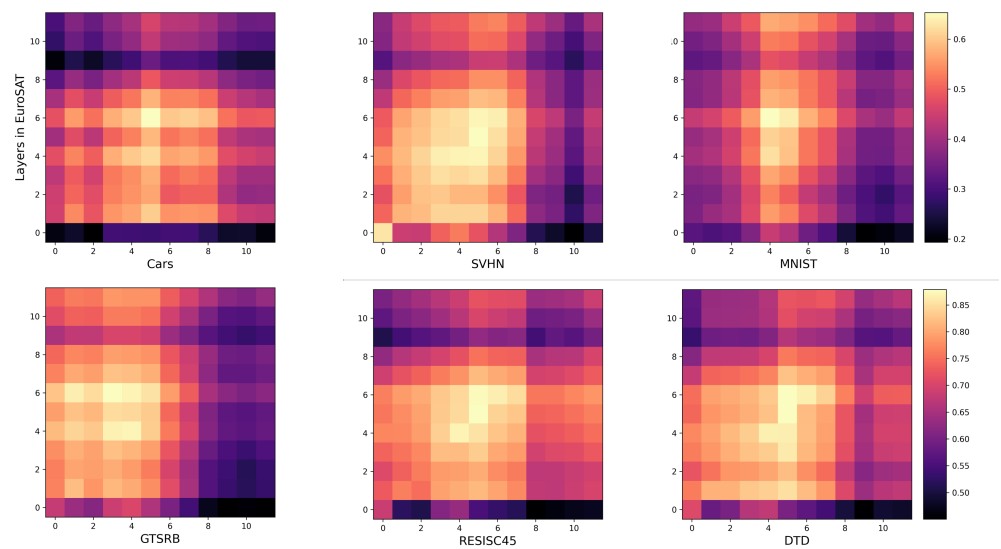

Figure 7: The CKA visualization results. Light colors indicate high similarity, and vice versa.

Layer-specific similarity of each model is shown in Figure 8.

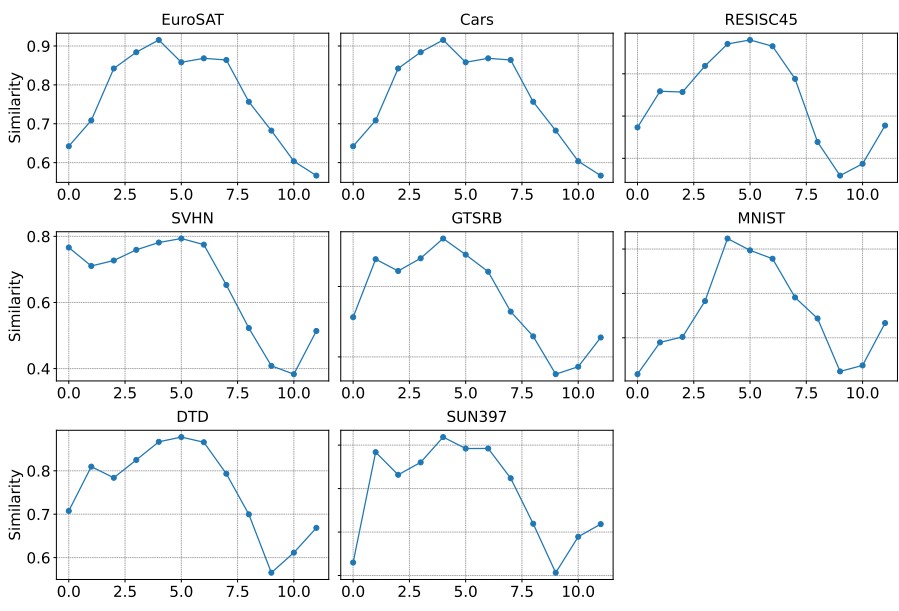

Figure 8: Layer-specific similarity analysis.

## B EFFICIENCY

Efficiency analysis based on ViT-L/14 model is given:

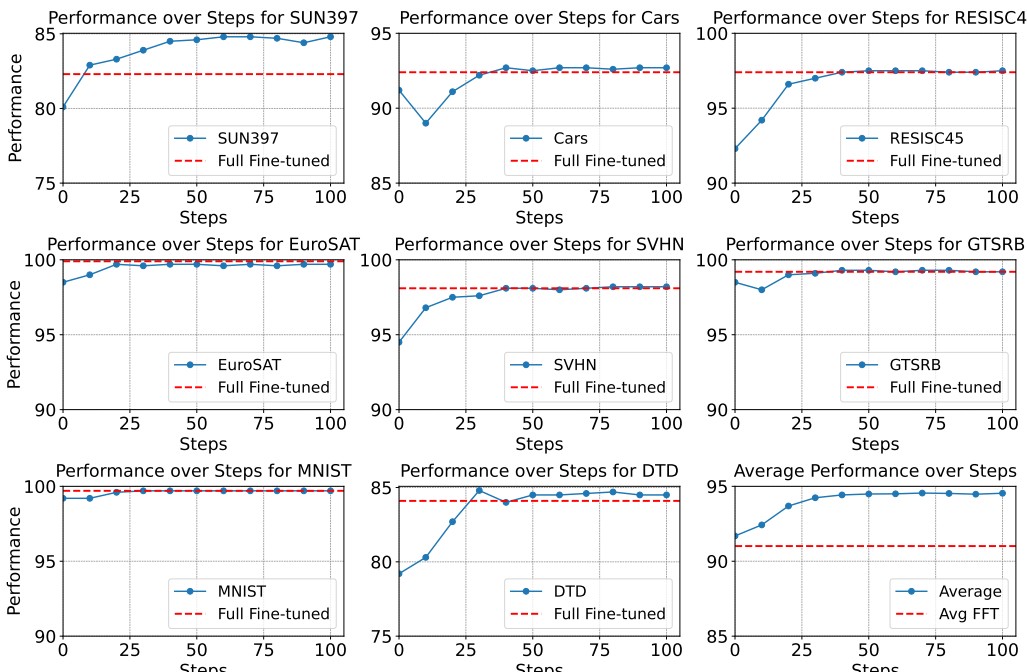

Figure 9: Performance versus steps compared to full fine-tuning based on ViT-L/14. Each red dashed line signifies the performance of full fine-tuning.

