# OpenReview forum: "Model-Driven Labeled Data Free Fine-tuning"
_ICLR.cc/2025/Conference — ICLR 2025 Conference Withdrawn Submission_

### Official Review · Reviewer_AmvL · 2024-11-02

**Soundness:** 2
**Presentation:** 1
**Contribution:** 2
**Rating:** 3
**Confidence:** 5

**Summary:**

The authors propose an unsupervised fine-tuning method based on meta-learning. They set up the meta-learning model as a weighted aggregation of the weights from different task networks and employ an innovative loss function to update the meta-learning model. During application, only a few steps of fine-tuning are required to adapt it to a specific task. This method does not require any labeled data and achieves better results compared to similar approaches.

**Strengths:**

- This method is an unsupervised approach that does not have the limitation of requiring labeled data.
- It requires only a small amount of training for application to downstream tasks.
- It overcomes the shortcomings of traditional EM loss.

**Weaknesses:**

- The writing of the paper has significant issues, including but not limited to the completeness of the sentence on line 219, the overly abstract nature of Figure 1, the formatting of Figure 5, and the symbols used in Algorithm 1.
- Some details are not clearly explained; for instance, I could not find what kind of \(\Phi\) the authors used to transform the weight residuals, nor whether they employed the task-specific or layer-specific approach in their experiments.
- The authors' main contribution is that the proposed method is unsupervised; however, it appears they merely replaced the supervised loss in the learning approach with an unsupervised loss without significant innovation compared to supervised meta-learning algorithms.
- The usability of the method seems limited. In practical applications. When focusing on solving a specific task, we typically do not collect data from other tasks. If the emphasis is on the task generalization capability of meta-learning, the authors have not provided any cross-task experiments to demonstrate whether the meta-learning model trained on these eight tasks can be applied to other datasets.

**Questions:**

- The authors' ablation study appears somewhat unusual. I believe it should only present the differences between the standard EM method, task-specific and layer-specific approaches, along with the hyperparameter differences in the sample filtering strategy. What is the purpose of this ablation study?
- What is the objective of the meta-learning model? Is it aiming to learn a weight representation that is similar across all tasks? In Equations 2 and 3, the first term aggregates all tasks, while the second term represents the information entropy of the task. However, in line 12 of Algorithm 1, all $L_i$ are summed. Is this equivalent to directly aggregating the first and second terms for all tasks? In other words, does the meta-network output for all tasks resemble that of the original network, while also exhibiting lower information entropy across all tasks?
- The classification layers for different tasks seem to vary. How is it possible to aggregate the weights in this case?

---

### Official Review · Reviewer_mHyu · 2024-11-03

**Soundness:** 2
**Presentation:** 2
**Contribution:** 3
**Rating:** 3
**Confidence:** 3

**Summary:**

This paper proposes a new approach for unsupervised fine-tuning. Traditional supervised fine-tuning approaches usually require extensive labeled datasets. This work introduces a model-driven technique that uses the combined capabilities of a pool of ready-to-use fine-tuned models. The approach is made of two steps:

1. Meta-Model Construction: By synthesizing a general "meta-model" from the collective knowledge of a model pool, the technique applies an entropy-based unsupervised learning method. This aims to reduce issues related to entropy minimization, like error accumulation, etc.

2. Model Specialization: The meta-model undergoes few-shot fine-tuning to align its representations with task-specific models, thereby achieving specialization without labeled data.

**Strengths:**

1. The motivation is clear. The paper is motivated by the challenge of limited labeled data in many domains, such as medical imaging and rare object detection. By proposing a method that fine-tunes models without labeled data, it offers a practical solution to overcome this bottleneck, making model adaptation feasible in low-data environments.

2. The problem is critical and important, and the idea looks novel.

3. The experiments use multiple datasets and model configurations (ViT-B/32 and ViT-L/14), and the results support the claims made regarding performance improvements.

**Weaknesses:**

1. The Theoretical justification for the proposed EM-based loss function is limited. Providing deeper theoretical insights or proofs showing why this particular entropy minimization method reduces error accumulation could strengthen the paper.

2. The analysis of task-specific and layer-specific approaches is insufficient. There is minimal discussion on when one might be more appropriate or effective than the other.

3. Lack of comparison with recent unsupervised and self-supervised fine-tuning methods. While the paper compares the proposed approach to full-parameter fine-tuning and multi-task learning baselines, it lacks comparisons with recent unsupervised or self-supervised fine-tuning methods.

4. It might be hard to reproduce the implementation. It's unclear how the hyperparameters are selected or optimized.

5. The presentation is not good. Some examples:
  - Mathematical notations: Eqn (4) and (5) are not clear and there's some inconsistent notations, such as $H_o(k)$ and $H_0^K$, etc.
  - Most paper citations: most papers should be cited like "(Niu et al. 2022)" instead of "Niu et al. (2022)".
  - Line 219: the reference "2" looks wrong.
  - Algorithm 1: the explanation above the algorithm and the algorithm are not consistent, such as some inputs explained are not listed in the algorithm input.

**Questions:**

1. Why does the proposed approach work? Just like weakness #1, it's necessary to have an analysis for this to strengthen the paper.

2. Why does the proposed approach outperform full supervised fine-tuning? Intuitively, it's highly possible that unsupervised approaches underperform supervised ones, although there might be some other advantages. If the unsupervised way is better, does that mean the label information is useless?

3. What's the available fine tuning models in the pool? Although you provide a link, it's necessary to introduce this clearly.

---

### Official Review · Reviewer_DhHm · 2024-11-03

**Soundness:** 1
**Presentation:** 1
**Contribution:** 1
**Rating:** 3
**Confidence:** 4

**Summary:**

This paper attempts unsupervised fine-tuning, which seeks to construct a model for a new task without leveraging any labeled data (as per the authors’ definition). To achieve said goal, the authors propose to first construct a meta-model from a collection of fine-tuned models and then specialize the meta-model for a task of interest via modifying its internal representation. Experiments are conducted on a series of image classification tasks to show the efficacy of the approach.

**Strengths:**

[S1] Constructing a meta-model for downstream specialization by mixing different model weights is interesting.

**Weaknesses:**

[W1] Unclear problem setup: It is unclear to the reviewer what task the authors are trying to solve (or specifically what counts as unsupervised fine-tuning.). First off, the authors never clearly mentioned how the new task of interest is specified and how different the task of interest should be compared to the tasks used for constructing the fine-tuned models (e.g., are the task of interest novel/unseen? Are the classes of the new task of interest relevant to the fine-tuning tasks in any manner?). From lines 271-274, the task of interest seems to be specified through one of the fine-tuned models. If that is the case, there is nothing unsupervised about the approach/setup since the supervision comes from a model trained on many labeled examples. Besides, if one can already fine-tune a model with labeled data, it is unclear why any practitioner would want to go through all the processes proposed by the authors to get a different model. Besides, it is unclear what the constraint of the problem setup is (e.g., lack of computes? No access to data due to privacy concerns?). It is difficult for the reviewer to know where the true contribution comes from without clearly understanding the problem's constraints.

[W2] Unclear reason for gains: While the authors show some performance improvement, it is unclear where the gains come from. Do they come from using multiple fine-tuned models? If yes, why would the proposed approach be appropriate for extracting relevant information from the multiple fine-tuned models?

**Questions:**

Suggestion:
[S1] The reviewer suggests having a section clearly spelling out the problem setup before even introducing the approach. It seems that the authors are attempting a slightly unusual problem setup, so it is worth spending time defining the setup and explaining the relevance/practicality of the problem setup.

---

### Official Review · Reviewer_7c28 · 2024-11-04

**Soundness:** 1
**Presentation:** 1
**Contribution:** 1
**Rating:** 1
**Confidence:** 5

**Summary:**

The paper presents a novel model-driven finetuning method as an alternative to supervised fine-tuning. Supervised finetuning relies on extensive labeled data, but this new method doesn't. It uses a diverse model pool's collective intelligence in a two-phase process. First, a general meta-model is created from the models' expertise. Then, this meta-model guides the iterative fine-tuning of the original models. Experimental results show that this approach outperforms full - parameter fine-tuning models and doesn't require supervised training. It's a cost-effective and scalable solution for data scarcity, making fine-tuning more practical in limited or no labeled data scenarios, and is a significant step for future research in unsupervised model enhancement.

**Strengths:**

This paper presents novel data pruning methods with some experimental demonstrations and intuitive explanations. It is easy to follow.

**Weaknesses:**

This is a very low-quality submission. When one starts to review it, the flaws are immediately apparent. The presentation gives an impression of carelessness and lack of attention to detail. The method proposed in this paper lacks the necessary theoretical and empirical support.

The submission contains many typesetting issues and typos. The text seems to have been hastily assembled without proper proofreading. Words are misspelled, punctuation is misplaced, and the formatting is inconsistent. This not only makes it difficult to read but also reflects poorly on the author's professionalism.

Comparisons with almost all the baselines (see the reference) in the Data-pruning field are absent. Baseline comparisons are crucial in research as they provide a reference point to evaluate the proposed method. Without these, it's impossible to determine the true value and effectiveness of the work in the context of existing techniques.

Furthermore, experiments were only carried out on a very simple and tiny-scale dataset. A limited dataset means that the results might not be generalizable. It fails to account for the complexity and diversity that a real - world application would encounter. Simple datasets often lack the nuances and challenges that more comprehensive ones possess.

However, there is no necessity for pruning on this toy - sets at all. Pruning is a technique designed to optimize and improve efficiency. But in the case of these simplistic toy sets, the potential benefits of pruning are negligible.


[1]. Moderate Coreset: A Universal Method of Data Selection for Real-world Data-efficient Deep Learning. ICLR-2023.

[2]. Coverage-centric Coreset Selection for High Pruning Rates. ICLR-2023.

[3]. Data Pruning via Moving-one-Sample-out. NeurIPS-2023.

[4]. Beating power law scaling via data pruning. NeurIPS-2022.

[5]. Mind the Boundary: Coreset Selection via Reconstructing the Decision Boundary. ICML-2024.

[6]. Dataset Pruning: Reducing Training Data by Examining Generalization Influence. ICLR-2023.

**Questions:**

See the weakness part.

---

### Note · Authors · 2024-11-20

I have read and agree with the venue's withdrawal policy on behalf of myself and my co-authors.